# State-of-the-Art Capability of Convolutional Neural Networks to Distinguish the Signal in the Ionosphere

**DOI:** 10.3390/s22072758

**Published:** 2022-04-02

**Authors:** Yu-Chi Chang, Chia-Hsien Lin, Alexei V. Dmitriev, Mon-Chai Hsieh, Hao-Wei Hsu, Yu-Ciang Lin, Merlin M. Mendoza, Guan-Han Huang, Lung-Chih Tsai, Yung-Hui Li, Enkhtuya Tsogtbaatar

**Affiliations:** 1Department of Space Science and Engineering, National Central University, Taoyuan City 320317, Taiwan; jason0010125@g.ncu.edu.tw (Y.-C.C.); chlin@jupiter.ss.ncu.edu.tw (C.-H.L.); davidhsieh@g.ncu.edu.tw (M.-C.H.); willy108623016@g.ncu.edu.tw (H.-W.H.); alterjohnnylife@g.ncu.edu.tw (Y.-C.L.); mmmendoza@g.ncu.edu.tw (M.M.M.); enter468@g.ncu.edu.tw (G.-H.H.); 2Skobeltsyn Institute of Nuclear Physics, Lomonosov Moscow State University, 119899 Moscow, Russia; 3Center for Space and Remote Sensing Research, National Central University, Taoyuan City 320317, Taiwan; lctsai@csrsr.ncu.edu.tw; 4AI Research Center, Hon Hai Research Institute, Taipei 114699, Taiwan; yunghui.li@foxconn.com; 5Department of Computer Science and Information Engineering, National Central University, Taoyuan City 320317, Taiwan; enkhtuya@g.ncu.edu.tw

**Keywords:** ionospheric sounding, artificial intelligence, image segmentation

## Abstract

Recovering and distinguishing different ionospheric layers and signals usually requires slow and complicated procedures. In this work, we construct and train five convolutional neural network (CNN) models: DeepLab, fully convolutional DenseNet24 (FC-DenseNet24), deep watershed transform (DWT), Mask R-CNN, and spatial attention-UNet (SA-UNet) for the recovery of ionograms. The performance of the models is evaluated by intersection over union (IoU). We collect and manually label 6131 ionograms, which are acquired from a low-latitude ionosonde in Taiwan. These ionograms are contaminated by strong quasi-static noise, with an average signal-to-noise ratio (SNR) equal to 1.4. Applying the five models to these noisy ionograms, we show that the models can recover useful signals with IoU > 0.6. The highest accuracy is achieved by SA-UNet. For signals with less than 15% of samples in the data set, they can be recovered by Mask R-CNN to some degree (IoU > 0.2). In addition to the number of samples, we identify and examine the effects of three factors: (1) SNR, (2) shape of signal, (3) overlapping of signals on the recovery accuracy of different models. Our results indicate that FC-DenseNet24, DWT, Mask R-CNN and SA-UNet are capable of identifying signals from very noisy ionograms (SNR < 1.4), overlapping signals can be well identified by DWT, Mask R-CNN and SA-UNet, and that more elongated signals are better identified by all models.

## 1. Introduction

The Ionosonde is an instrument used for monitoring the ionosphere by reflecting the radio wave in different frequencies. The ionospheric signals are characterized by the frequencies of the radio waves they reflect, and the traveling time of the reflected radio wave is used to estimate the virtual height of the signal. Active sounding allows the determination of important local ionospheric parameters for regional modeling of the ionosphere and the calculation of ionospheric indices. The oblique sounding with sweeping frequency allows the measurement of the vertical ionization profile above the sea surface [1].

It is important to accurately identify each signal layer because it contains different physical properties and mechanisms that need to be determined, respectively. For instance, the ionospheric anomaly such as equatorial sporadic E-layer (Es) can be produced by two-stream plasma instability [2], and equatorial F-layer ionospheric irregularities can be caused by depletion within the bubbles [3]. These phenomena affect radio propagation and communication. More precise classifications are required for further understanding and monitoring of the space weather.

An ionogram is a representation of ionosonde data. It shows the vertical profiles of the ionospheric signals as a function of frequency. The ionograms used in this study are produced by Vertical Incidence Pulsed Ionospheric Radar (VIPIR) [4], which is a digital ionosonde consisting of several antennas that can produce thousands of ionograms per day. The amount of raw data from two VIPIRs and two dynasondes is estimated to be about 20 Terabytes (TB) per year.

Due to the large amount of data generated by modern ionosondes and the demand for real-time information of ionosphere conditions, manual identification of each layer for each ionogram needs to be replaced by automatic detection [1]. Tsai and Berkey (2000) [5] developed a fuzzy segmentation and connectedness technique to classify different signals and determine their critical frequencies for the ionograms in the Taiwan region. They reported an 85% success rate of determining foF2, which is the critical frequency (i.e., maximum frequency of the reflected signal) of the F2 layer in the ionosphere, in the daytime and 50% in the nighttime ionograms. However, the method cannot accurately predict and classify signals: some noise is predicted as signals in every ionogram, and echoes cannot be distinguished from the primary signals.

More recently, different deep learning techniques based on convolutional neural network (CNN) have shown promising results for the recovery of useful signals in ionograms. De La Jara and Olivares (2021) [6] applied a convolutional encoder–decoder model to extract F-layer signals in the ionograms produced by the Jicamarca station in Peru. In this work, they do not distinguish different types of F-layer signals, and signals in other layers are not identified. They used intersection over union (IoU) to evaluate the performance of their model and reported an average IoU of 0.569. Mochalov et al., 2019 [7] applied a deep neural network (DNN) U-Net to identify E, F1, and F2 layers of mid-latitude ionosphere observed by the ionosonde “Parus-A” at Kamchatka. The three layers selected for identification in this work are strong and do not overlap with each other in their ionograms. They tested the performance of their DNN model on the ionograms compressed to two different sizes, 64 × 48 pixels, and 192 × 144 pixels, and reported lower accuracy for higher-resolution ionograms and vice versa. The highest accuracy reported in their study is for the F2 layer in the ionograms of 64 × 48 pixels, with Dice-Coefficient Loss (DCL) 0.10842, which is equivalent to IoU = 0.8. Xiao et al., 2020 [8] developed a deep-learning method for ionogram automatic scaling (DIAS) which is a U-shaped network consisting of an encoder and a decoder network and using a feature pyramid network (FPN) in place of the skip connections in a U-net. DIAS can be implemented with different backbone networks. The data used in the study are the mid-latitude ionograms produced by Wuhan DPS4D ionosonde. DIAS implemented with different backbone networks were applied to extract E, F1, and F2 layers, which are three distinct layers that do not overlap or cross each other in their ionograms. They reported a recall rate of 90.46% and an F-score of 93.05%.

While these earlier studies have successfully extracted distinct signals from ionograms, they have not explored the possibility of using CNN models to extract overlapping or highly contaminated signals. Taiwan region is under the equatorial ionization anomaly (EIA) region. The ionograms in such regions are much messier and more overlapped than the mid-latitude data. In order to distinguish and extract two overlapping signals, the neural network model must be designed in such a way that multiple signal labels can be assigned to a single pixel. In addition to the challenge of signals highly overlapping with each other, the ionograms in the Taiwan region contain strong stripes of man-made noise caused by radio stations or other transmitters. Such noise often overlaps useful ionospheric signals and obscures their critical frequencies, causing difficulties in identifying the true profiles of the signals [1].

In this work, the aim is to employ advanced algorithms of deep learning to automatically distinguish and classify all ionospheric layers from highly noisy ionograms. Specifically, we train and apply five artificial neural network models: DeepLab [9], fully convolutional DenseNet (FC-DenseNet) [1,10], deep watershed transform (DWT) [11,12], Mask R-CNN [13,14], and Spatial Attention U-Net (SA-UNet) [15,16]; all of these have been successfully applied to image segmentation [16,17,18,19,20,21,22,23,24,25]. It is important to note that all these models are available at https://www.kaggle.com/ (access on 29 March 2022) and thus can be easily applied and verified. We compare the results from each model to evaluate their respective strengths and weaknesses in identifying different signals under different ionospheric conditions and noise levels. Our results can help scientists to select the optimal model to identify specific ionospheric signals of their interest.

## 2. Data

The ionograms we use in this work were acquired from the VIPIR ionosonde located at Hualien, Taiwan (23.99° N, 131.61° E). The time resolution is 2 min [1], sweeping frequency is from 1 to 22 MHz, the virtual height range is from 0 to 800 km, and the amplitude of the signal is measured in the range of 1 to 100 dB. The dimension of the original ionograms is 2300 pixels in frequency and 1700 pixels in virtual height. A typical example of ionograms is shown in Figure 1. It can be seen that the ionogram contains not only signals but also noise and contamination. 

Except for during geomagnetic storm time, the useful signals usually only appear in a narrower range of frequencies and virtual heights. Therefore, to reduce the computing time, the ionograms are downsized to a frequency range from 1.58 to 20.25 MHz and virtual height range from 66 to 600 km. The resulting ionograms contain 800 pixels in frequency and 400 pixels in virtual height, as shown in Figure 2.

## 3. Method

### 3.1. Preparation of Ground Truth

The ground truths are manually labeled ionospheric signals. The ionospheric layers and the corresponding label names are listed in Table 1. We distinguish 11 labels (classes) for the models during the training. The manual labeling is performed by using a polygonal annotation tool, LabelMe (Wada, 2016), which is free and available on GitHub (https://github.com/wkentaro/labelme (accessed on 29 March 2022)), and has generally been used for labeling of polygon, rectangle, circle, line, and point. In our case, we use polygons to label the signals around the outer edge. Because of this method and the limitation of manual labeling, the ground truth would not be exactly equal to the real signals. Moreover, the signals that are too faint to be accurately identified by eye are not labeled.

We have labeled 6131 ionograms and used 3924 ionograms (≈64%) for training, 981 (≈16%) for validation, and 1226 (≈20%) for testing. The test set is used to test the performance of the model after the whole training. The validation set is used for supervising and calibrating the model during the training, and for the selection of hyperparameters to reduce underfitting and overfitting problems. Table 2 shows the percentage of sample numbers of each label in the training, validation and test sets. The percentage corresponds to the percentage of the ionograms that contain the indicated signal label. Since an ionogram can contain multiple labels and not all signal labels appear in all ionograms, the sum in each set is not 100%. 

### 3.2. Models

Five convolutional neural network (CNN) models used in this study and their operations are briefly described in the following sections. The total number of parameters and layers of the models are listed in Table 3. We provide our models and data set on Kaggle (https://www.kaggle.com/ (accessed on 29 March 2022)).

#### 3.2.1. DeepLab

The basic architecture of DeepLab [9] consists of three parts: (1) backbone, which is a Deep CNN (DCNN), either a VGG or ResNet structure, for feature extraction, (2) Spatial Pyramid Pooling, which consists of several parallel atrous convolution layers with different dilation rates, and (3) an upsampling part, which concatenates the output from (2) with corresponding levels in DCNN, and then applies two bilinear upsampling and a convolution layer to return the size of the output image to that of the input. In our study, we use ResNet-34 [26] as the backbone DCNN.

In a DeepLab network, ‘ReLU’ is used as the activation function throughout the model for better computational efficiency and to avoid the vanishing gradient problem. Unlike soft-max, the ReLU function does not normalize the output probability. Therefore, more than one label can be assigned to a same pixel, achieving the goal of multi-label classifications. The loss and IoU values of training and validation sets during training are monitored to avoid overfitting.

#### 3.2.2. FC-DenseNet

Fully Convolutional DenseNet (FC-DenseNet) is a Fully Convolutional Network (FCN) model evolved from a CNN model called DenseNet [27]. The original version of FC-DenseNet, FC-DenseNet103 [10], consists of 670 layers and more than 19 million parameters. Given that our data set consists of more than 6000 images of 800 × 400 pixels, the computing power and memory space required to apply FC-DenseNet103 significantly exceed the limitation of our computing environment, Kaggle. Therefore, we constructed our lighter-weight version, FC-DenseNet24, by gradually reducing the number of layers and parameters until the model can run on Kaggle. The total number of layers and parameters are listed in Table 3, and the detail of the architecture is described in [1]. In both FC-DenseNet103 and FC-DenseNet24, the soft-max function is used as the activation function for the convolutional layer of output. Since the soft-max function normalizes the output probability, i.e., the probabilities of all labels are summed up to 1, there is usually only one most probable label for each pixel. Therefore, at the locations where two signals are overlapping, FC-DenseNet24 is expected to predict only one, instead of both, signal label. To avoid overtraining or undertraining, the loss of validation set during the training is used for revising the learning rate.

#### 3.2.3. DWT

Watershed transform [11,12] is a mathematical method that can be applied to solve image segmentation problems. Bai et al., 2017 constructed a CNN model, called deep watershed transform (DWT), to compute the watershed transform using an Artificial Neural Network.

DWT model consists of two networks. The first one is called “Direction Network (DN)”, which learns the direction of descent of the watershed energy. The result of DN is subsequently passed to the second network called “Watershed Transform Network (WTN)”, which determines the properties of different isolated regions in the input image. The pooling architecture of DN is the same as VGG16 [28]. The only difference is that the inputs to VGG16 are RGB images, which require 3 channels. DN in the original DWT adds an additional channel for semantic segmentation data. 

In our application, we set the number of input channels to 1 because our inputs are grayscale images that require only one channel. To allow multi-label assignment to a same pixel, we change the activation function of the output layer to sigmoid. Since the output probabilities are not normalized, they can be used in multi-label classification (see Section 3.2.5 for more explanation). 

The IoU of the validation set during the training is monitored to avoid overfitting: if the IoU of the next epoch is lower than the previous epoch, we increase the batch size. The reason that we increase the batch size instead of decreasing the learning rate is that the former can reduce the training time [13].

#### 3.2.4. Mask R-CNN

Mask R-CNN [14] is a model that can produce three levels of outputs: bounding boxes of features, class of each bounding box, and mask (i.e., useful signal) of the feature in each bounding box. The accuracy of the mask depends on the accuracy of the bounding box.

The architecture of Mask R-CNN is illustrated in Figure 3. The model consists of three main parts: Backbone, which is composed of convolutional neural network and feature pyramids network (CNN + FPN), region proposal network (RPN), and region of interest align (ROI Align).

The Backbone (CNN + FPN) first generates feature maps of different spatial scales. RPN then identifies different ROIs in the feature maps. The ROIs are either regions containing a feature or regions near a feature. Lastly, ROI Align uses bilinear interpolation to achieve the equalized scale of different ROIs, and finally maps each ROI back to its corresponding feature map and determines the bounding boxes, classes, and masks. The process is as follows: the degree of overlap between the ROIs and ground-truth bounding boxes is first used to associate each ground-truth bounding box with one or several ROIs. One ROI can be associated with more than one ground-truth bounding box. At this stage, each ground-truth bounding box contains only the pixels (mask) of one ground truth (i.e., labeled signal). The ROIs are then deconvolved to produce the mask for each bounding box. In other words, instead of predicting the label(s) for a pixel, as in the other four models, Mask RCNN predicts the pixels (mask) associated with a label, and a pixel can be associated with different labels, thereby achieving the goal of multi-labeling.

In our application, we apply two strategies to avoid overfitting. One is to monitor the IoU of the validation set, and the other is to pre-train the detection of mask, class, and bounding boxes.

#### 3.2.5. SA-UNet

Spatial attention U-Net (SA-UNet) was developed by Guo et al., 2019 and Guo et al., 2021 [15,16], who have demonstrated their capability in identifying tiny or elongated features. The model (illustrated in Figure 4) has a U-Net architecture with five major components: the convolution block, maximum pooling, spatial attention module, transpose convolution layer and skip connection.

The convolution block is composed of a convolution layer followed by a DropBlock [29] layer and a batch normalization layer. The spatial attention module is placed at the bottleneck of U-Net and is designed to capture the “attention” of the feature space.

In our operation, four hyperparameters: epoch number, batch-size, kernel-size, and dropout probability are manually tuned, and the resulting IoUs of the validation set are compared to evaluate the performance. It has been shown that DropBlock can reduce overfitting and improve the accuracy of semantic segmentation [14,29]. 

To achieve multi-label classification, we use the sigmoid function in place of the soft-max function. Since the soft-max function normalizes the output probability, there can only be one most probable label for the same pixel. In contrast, the sigmoid function does not normalize the output. Therefore, there can be more than one highest probability label at a pixel. 

The original SA-UNet used 3 × 3 kernels in the convolutional blocks, and the DropRate was designed as ((1 − P)/(K^2^)) ∗ (H/(H − K + 1)) ∗ (W/(W − K + 1)), where K is the kernel size, H is the height of the image, W is the width of the image, and P is the keep probability. We changed the kernel size of the convolutional blocks to 7 × 7 to improve the overall performance, and we modified the DropRate to be (1 − P)/(K^2^) to reduce the computational complexity.

### 3.3. Evaluation of Performance

The performance of an image segmentation model in our study is evaluated by the intersection over union (IoU) [1] defined as follows:(1)$${\mathrm{IoU}}_{i,\;\;k}=\frac{I_{i,\;k}}{U_{i,\;k}}$$
(2)$${\mathrm{IoU}}_{i}=\frac{\sum_{k=1}^{K}I_{i,\;k}}{\sum_{k=1}^{K}U_{i,\;k}}$$
where IoU*_i,k_* is the IoU of signal label *i* in ionogram *k*, and IoU*_i_* is the mean IoU of signal label *i. I_i,k_* is the total number of intersecting pixels between prediction and ground truth for signal label *i* in ionogram *k*, and U*_i,k_* is the total number of union pixels for the same signal label in the same image. The number *k* is the total number of ionograms in the training, validation, or testing set.

In addition to comparing the IoU of each signal from different models, we also examine the effects of the following three factors on the performance of different models.

#### 3.3.1. Signal-to-Noise Ratio (SNR)

In the ionograms, the useful signals are contaminated by the background instrumental noise and the noise from man-made radio signals. To evaluate the level of the noise and the readability of the ionogram, we adopt the signal-to-noise ratio (SNR) defined by Mendoza et al., 2021 [1]. Mendoza et al., 2021 [1] introduced parameters CS, CS_n_, and CS_bg_, which are the characteristic amplitude of the signal, the characteristic amplitude of the man-made noise, and the characteristic amplitude of the background, respectively, in the statistical distribution of the pixel intensities of an ionogram. CS_bg_ is defined as the median intensity of all pixels which are less than the most probable intensity I_mp_ in the statistical distribution of pixel intensity: ${\mathrm{CS}}_{\mathrm{bg}}=\mathrm{Median}(I<I_{\mathrm{mp}})$, CS_n_ is defined as the median intensity of all the pixels which are higher than CS_bg_: ${\mathrm{CS}}_{n}=\mathrm{Median}\left(I>{\mathrm{CS}}_{\mathrm{bg}}\right)$, CS is defined as the median intensity of all the pixels in the statistical distribution of the labeled signal’s pixel amplitude: $\mathrm{CS}=\mathrm{Median}\left(I_{s}\right)$. The SNR is defined simply as the ratio between the characteristic signal (CS) and the characteristic man-made noise (CS_n_):(3)$$\mathrm{SNR}\equiv \frac{\mathrm{CS}}{{\mathrm{CS}}_{n}}$$

#### 3.3.2. Circumference-over-Area Ratio (C/A)

Since the labeling of the signals is along the outer edge of the signal, the labeled area (i.e., the ground truth) is slightly larger than the area of the true signal. Since our neural network models are expected to predict the true signal, the difference can affect the apparent performance of the models [1]. It is shown that the difference between the labeled and true signal areas is related to the shape of the signal, and it can be defined by a shape parameter: circumference-over-area ratio (C/A). The larger the C/A, the smaller the difference. We will examine how C/A would affect the performance of different models.

#### 3.3.3. Overlapping Ratio (OR)

The level of overlapping between two signals is calculated by the overlapping ratio (OR), which is defined as the intersection area of two signals over the total area of the signal of interest. For instance, the OR of signal_1_ by signal_2_ is calculated as
(4)$$\mathrm{OR}\equiv \frac{{\mathrm{signal}}_{1}\cap {\mathrm{signal}}_{2}}{{\mathrm{signal}}_{1}}$$
where signal_1_ is the area of the signal of interest and signal_2_ is the area of another signal, which intersects the signal of interest. In this work, we focus on the overlapping situation of the F2 layer because Fbo and Fbx usually have larger overlapping areas than other layers. 

Figure 5 shows an example of the effect of overlapping. Panel (a) is the original ionogram which contains the signals Fbo and Fbx, and panel (b) is the ground truth, i.e., the labeled signals. The overlapping region is marked in red. Fbo and Fbx identified by the model DeepLab are plotted in panels (c) and (d), and those identified by the model FC-DenseNet24 in panels (e) and (f). Panels (c) and (d) show that identified Fbo and Fbx are both continuous with almost no gaps, which means that most of the pixels in the overlapping region are correctly identified as both Fbo and Fbx by DeepLab. In contrast, a wide gap can be seen in panel (f). This indicates that the model FC-DenseNet24, implemented with soft-max function at the output layer, is less capable of identifying the two signals in their overlapping region. Further, we will examine whether and by how much OR would affect the prediction accuracy of different models.

## 4. Results and Discussion

All of our models are written using Python 3.7.6 with NumPy 1.18.5, TensorFlow 2.3.0, and Keras 2.4.3, and the computations are conducted on Kaggle using two NVIDIA TESLA P100 GPUs and 16 GB of RAM. The training and inference times of different models are listed in Table 4. It can be seen that the inference time is very short, which makes it possible to apply the models for automatic processing of the ionograms in real time.

In our study, IoU < 0.2 is considered as “not identified” and IoU > 0.6 as “well identified”. In the following sections, we first compare the performance (IoU) of different models in recovering different ionospheric signals and then examine the effects of SNR, C/A, and OR on their performance. As one can see in Table 5, only three classes are predicted by all the models with high accuracy of IoU > 0.6: Es, Fbo, and Fbx. In the next subsections, we will use these three classes to analyze how various effects influence the accuracy of the five models considered.

### 4.1. Comparison of Model Performance

Table 5 shows the accuracy (IoU) of identifying each signal label by each model. The table shows that Es, Fbo, and Fbx are the best-identified signals. All five models can identify them with IoU > 0.5. This is because they are usually prominent and more accurately labeled, and the percentages of these signals in our data set are high (cf. Table 2). In contrast, most models cannot identify Eo, Ex, Esx, Fco, and Fcx. This is because the percentages of these signals are low, with Fco and Fcx less than 1% and Eo and Ex less than 10%. They are also often small and faint causing inaccuracy in labeling them. 

The three well-represented, prominent signals, Es, Fbo, and Fbx, are best identified by SA-UNet with IoU > 0.7. However, SA-UNet cannot identify Eo, Ex, Eso, Esx, Fax, Fco, and Fcx. Mask R-CNN, while not the best model for the prominent signals, can identify all these faints or poorly represented signals to some degree (cf. Table 2). Most models can identify Fao with moderate IoU, with FC-DenseNet achieving the highest IoU.

As a comparison, the Dice-Coefficient Loss (DCL) values achieved by Mochalov and Mochalova (2019) [7] are 0.18859, 0.22881, and 0.22209 for F2, F1, and E layers, respectively, for ionograms compressed to a size of 192 × 144 pixels. These values can be converted to IoU by Equations (5) and (6): (5)$$\mathrm{DCL}=1-\frac{2\left|X\cap Y\right|}{\left|X\right|+\left|Y\right|}$$
(6)$$\mathrm{IoU}=\frac{\left|X\cap Y\right|}{\left|X\cup Y\right|}=\frac{\left|X\cap Y\right|}{\left|X\right|+\left|Y\right|-\left|X\cap Y\right|}=\frac{\frac{\left|X\cap Y\right|}{\left|X\right|+\left|Y\right|}}{1-\frac{\left|X\cap Y\right|}{\left|X\right|+\left|Y\right|}}=\frac{1-\left(1-\frac{2\left|X\cap Y\right|}{\left|X\right|+\left|Y\right|}\right)}{1+\left(1-\frac{2\left|X\cap Y\right|}{\left|X\right|+\left|Y\right|}\right)}=\frac{1-\mathrm{DCL}}{1+\mathrm{DCL}}$$
where *X* is ground truth and *Y* is the prediction by models.

The resulting IoUs for the aforementioned DCLs are IoU = 0.68 (F2), 0.63 (F1), and 0.64 (E), respectively. The best IoUs achieved by our models are 0.745, 0.519, and 0.722 for Fbo, Fao, and Es to a size of 400 × 800 pixels. Our models achieve better accuracy for Fbo and Es layers even though our ionograms are 10 times larger. In Xiao et al. (2020) [8], they reported a best overall recall rate of 0.9046 and the best F-score (=1 − DCL) of 0.9305. Based on the same calculation, the recall rate and F-score achieved by our SA-UNet model are 0.982 and 0.9339. The best predictions of F2 from Mochalov and Mochalova [7] and Xiao et al., [8] are compared with our best prediction in Table 6.

### 4.2. The Effect of Signal-to-Noise Ratio

To demonstrate the effect of SNR on the performance of each model, we use Fbo, the most populated signal in our data set, as an example. The first five panels of Figure 6 show the scatter plots of IoU*_i,k_* vs. SNR of Fbo. Each point represents the IoU and SNR of Fbo in a single ionogram *k*, and the overplotted solid lines represent the median IoU of the points at each SNR. The median IoU profiles of different models are compared in the sixth panel. These panels indicate a positive correlation between IoU and SNR, as expected. They also show that, except for DeepLab, the median IoU of our models is higher than 0.6 even at very low SNR. This indicates that four out of our five models are capable of extracting useful signals from noisy ionograms.

### 4.3. The Effect of Circumference-over-Area Ratio (C/A)

To demonstrate the effects of the C/A ratio on model performance, we show one example of an elongated signal (Fbo: C/A~7.5 to 22) in Figure 7 and one example of a compact signal (Es: C/A~4 to 16) in Figure 8. The format is the same as that in Figure 6: the first five panels show the scatter plots of IoU*_i,k_* vs. C/A with median IoU overplotted as solid lines, and the median IoU of all models are compared in the sixth panel.

The figures reveal a positive correlation between IoU and C/A, especially when C/A is low. For most models, their median IoU of both Fbo and Es become higher than 0.6 and less dependent on C/A once C/A is higher than 8. The only exception is model DeepLab. While its median IoU of Es becomes higher than 0.6 for C/A > 8 (Figure 8), its median IoU of Fbo does not exceed 0.6 even for C/A > 15 (Figure 7).

Overall, the figures indicate that the IoU of more elongated signals is higher than that of more compact signals.

### 4.4. The Effect of Overlapping Ratio (OR)

The effect of the overlapping ratio on the identification of Fbo and Fbx by different models is shown in Figure 9 and Figure 10, respectively. The format is the same as in Figure 6: the scatter plots of IoU*_i,k_* vs. OR are shown in the first five panels and the median IoU of different models are compared in the sixth panel.

The figures show that, except for DeepLab and FC-DenseNet24, the median IoU of other models are higher than 0.6 for both Fbo and Fbx regardless of the overlapping ratio, indicating that SA-UNet, DWT, and Mask R-CNN can well-identify both Fbo and Fbx irrespective of how much the two overlap.

As explained earlier, the reason that FC-DenseNet cannot correctly identify two overlapping signals is that the soft-max function is implemented as an activation function in the output layer. As a result, it is impossible to assign multiple labels to the same pixel. In this case, if a pixel is both Fbo and Fbx, the label with higher weighting or higher sample percentage in the data set is more likely to have a higher probability and be assigned to the pixel.

## 5. Conclusions

Different ionospheric signals can be used to infer important physical properties in different layers of the ionosphere. Therefore, it is important to accurately identify and distinguish them. However, some of these signals often overlap with each other and can be very faint. A further challenge for identifying the signals in the ionograms acquired in the Taiwan region is the strong man-made radio transmission contaminating useful ionospheric signals. 

In this work, we train five deep learning CNN models, SA-UNet, DWT, Mask R-CNN, DeepLab, and FC-DenseNet24, to classify the ionospheric signals, and compare their performance. Our results show that all five models can identify prominent signals or signals with a high sample percentage in the data set, and SA-UNet is the best model to identify such signals, achieving IoU > 0.7. For signals with too low sample percentage and/or signals too faint to be identified by all other models, Mask R-CNN can be applied to achieve a low to moderate IoU of approximately 0.2 to 0.4.

The examination of the effects of signal-to-noise ratio (SNR), circumference-to-area ratio (C/A), and overlapping ratio (OR) on the model performance indicate that SA-UNet, DWT, Mask R-CNN, and FC-DenseNet24 can achieve median IoU > 0.6 even for very noisy ionograms (SNR < 1.4). All models show a positive correlation with C/A for compact signals (C/A < 8), and, except for DeepLab, become less affected by C/A for more elongated signals (C/A > 8). SA-UNet, DWT, and Mask R-CNN can correctly identify and distinguish overlapping signals, with a median IoU higher than 0.6, regardless of the overlapping ratio. Due to the design of FC-DenseNet24, it cannot assign multiple labels to one image pixel. As a result, it can only identify the stronger or better represented one among the overlapped signals.

We have shown that the CNN models presented here provide high accuracy for three classes: Es, Fbo, and Fbx (see Table 5). The classes (Eo, Ex, Eso, Esx, Fco, and Fcx) that are poorly identified, or unidentified, by our models are those with less than 15% samples in our data set. The insufficiency of the training data causes the models undertrained to identify these classes. Given that the model (Mask R-CNN) that performs best on these poorly represented labels is also the one implemented with the highest number of layers and parameters among all five models, we believe that a more advanced network with deeper learning [23,24,25] will give better accuracy in the segmentation of the ionograms. The development of advanced models will be a subject of our future work.

## Figures and Tables

**Figure 1 sensors-22-02758-f001:**
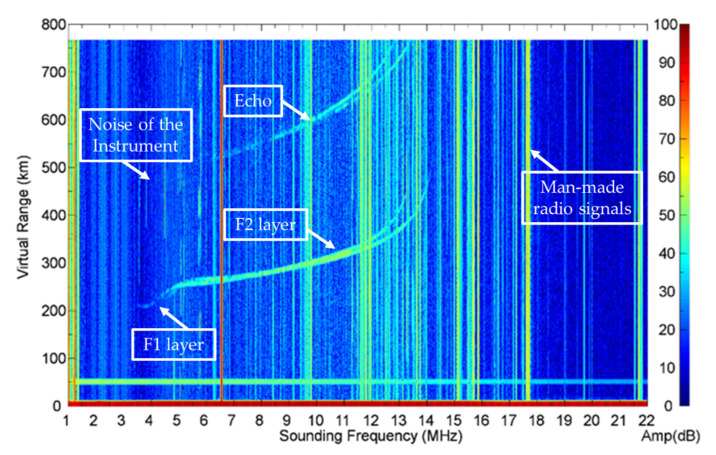
An example of an original ionogram.

**Figure 2 sensors-22-02758-f002:**
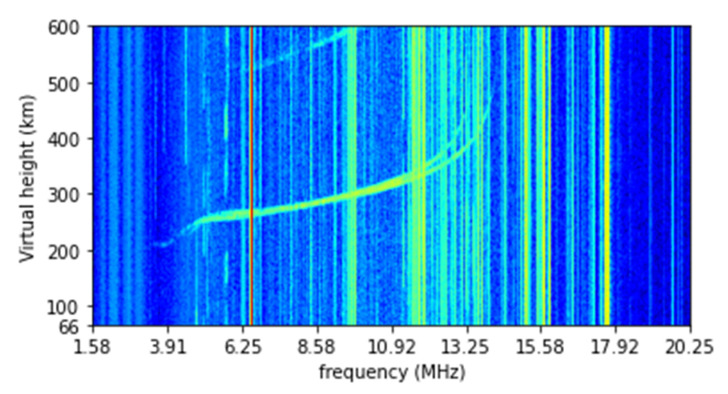
Downsized ionogram image.

**Figure 3 sensors-22-02758-f003:**
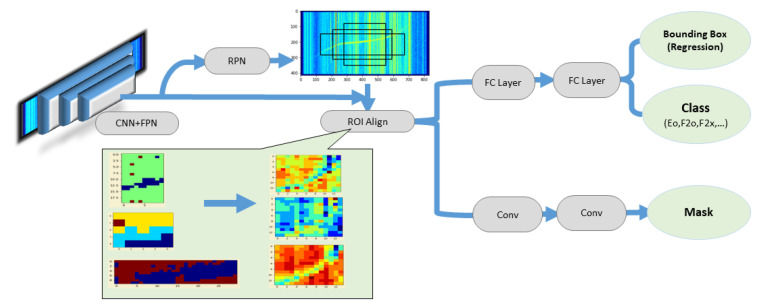
The architecture of Mask R-CNN.

**Figure 4 sensors-22-02758-f004:**
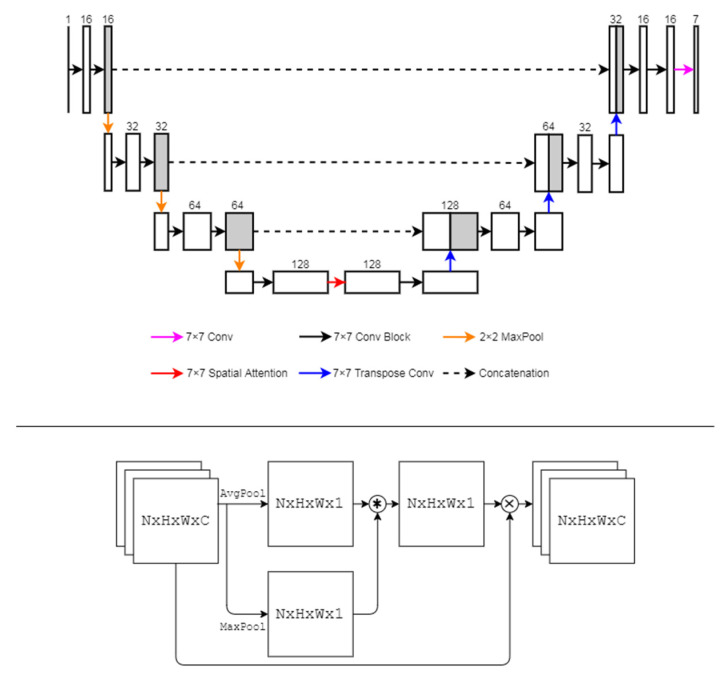
The architecture of SA-UNet.

**Figure 5 sensors-22-02758-f005:**
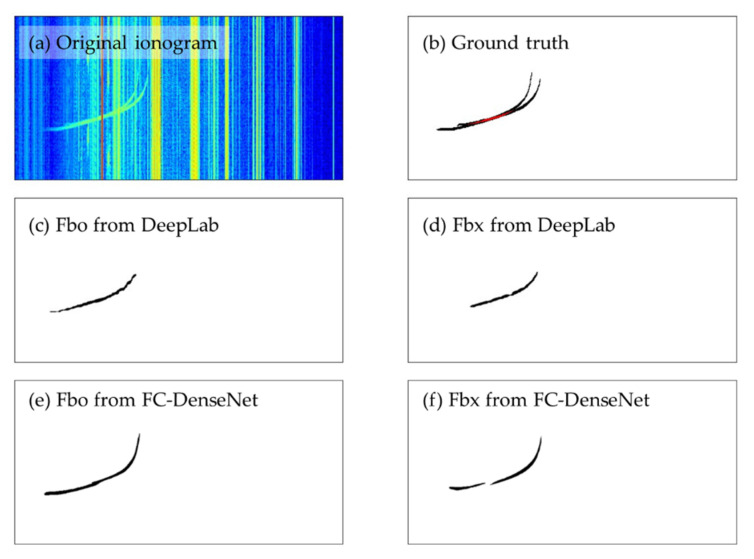
An example of the overlapping ordinary and extraordinary signals from the F2 layer: (**a**) the original ionogram; (**b**) the ground truth labeled in the original ionogram (black pixels are the non-overlapping part and red pixels the overlapping part of the signals); (**c**,**d**) the output of DeepLab for Fbo and Fbx, respectively; (**e**,**f**) the output of FC-DenseNet for Fbo and Fbx, respectively.

**Figure 6 sensors-22-02758-f006:**
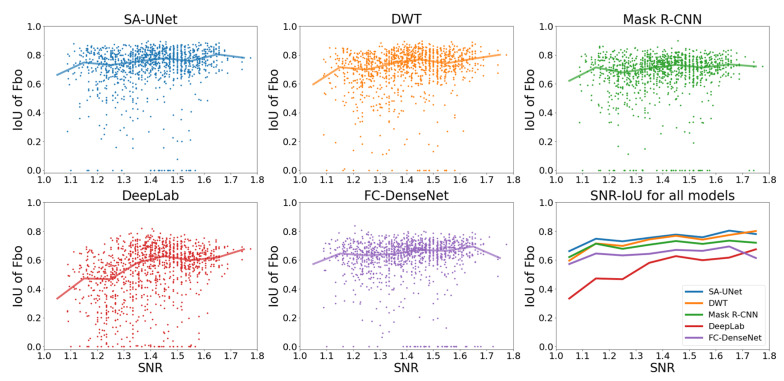
Comparison of IoU of Fbo vs. SNR for all models (different colors represent different models: blue: SA-UNet, orange: DWT, green: Mask R-CNN, red: DeepLab, purple: FC-DenseNet).

**Figure 7 sensors-22-02758-f007:**
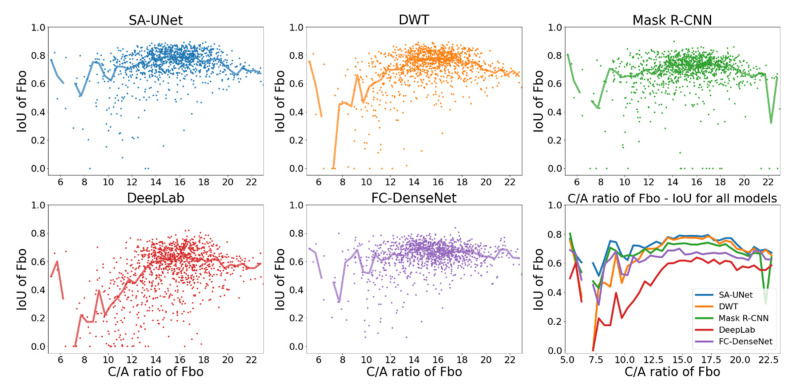
Comparison of IoU of Fbo vs. C/A ratio for all models (different colors represent different models: blue: SA-UNet, orange: DWT, green: Mask R-CNN, red: DeepLab, purple: FC-DenseNet).

**Figure 8 sensors-22-02758-f008:**
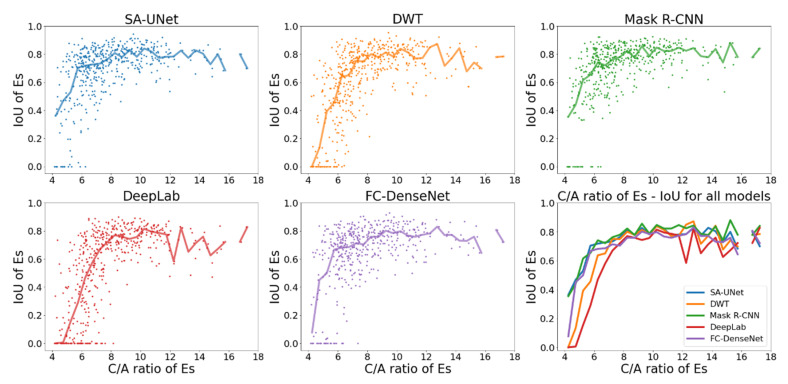
Comparison of IoU of Es vs. C/A ratio for all models (different colors represent different models: blue: SA-UNet, orange: DWT, green: Mask R-CNN, red: DeepLab, purple: FC-DenseNet).

**Figure 9 sensors-22-02758-f009:**
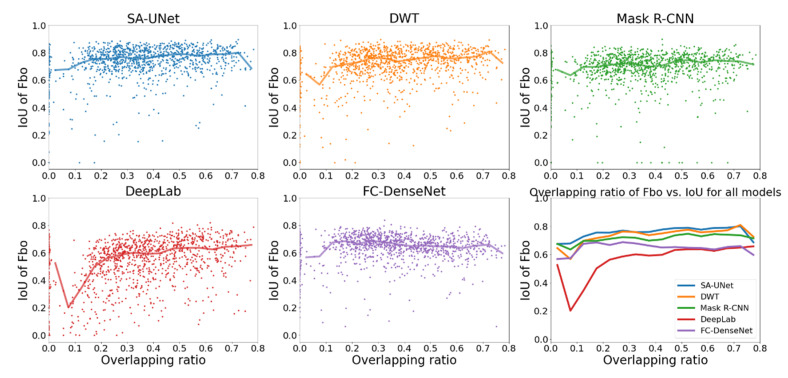
Comparison of IoU of Fbo vs. OR for all models (different colors represent different models: blue: SA-UNet, orange: DWT, green: Mask R-CNN, red: DeepLab, purple: FC-DenseNet).

**Figure 10 sensors-22-02758-f010:**
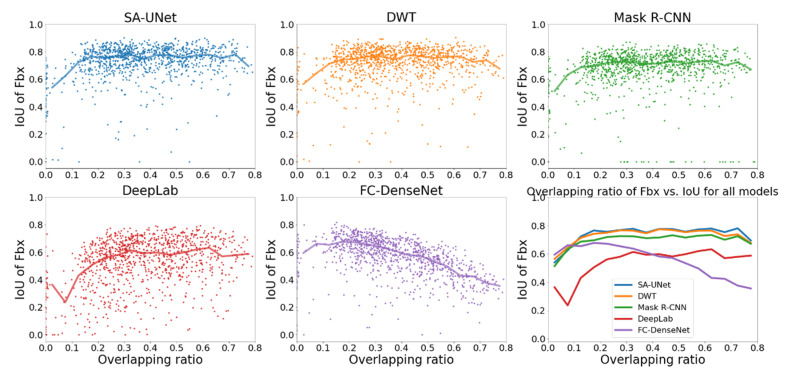
Comparison of IoU of Fbx vs. OR for all models (different colors represent different models: blue: SA-UNet, orange: DWT, green: Mask R-CNN, red: DeepLab, purple: FC-DenseNet).

**Table 1 sensors-22-02758-t001:** Ionospheric layers with their corresponding label names.

Ionospheric Layer	Class Label
E layer ordinary mode	Eo
E layer extraordinary mode	Ex
Sporadic E layer ordinary mode	Eso
Sporadic E layer extraordinary mode	Esx
Sporadic E layer	Es
F1 layer ordinary mode	Fao
F1 layer extraordinary mode	Fax
F2 layer ordinary mode	Fbo
F2 layer extraordinary mode	Fbx
F3 layer ordinary mode	Fco
F3 layer extraordinary mode	Fcx

**Table 2 sensors-22-02758-t002:** Percentage of each label for three different sets.

	Eo	Ex	Es	Eso	Esx	Fao	Fax	Fbo	Fbx	Fco	Fcx
Training	7.6	1.1	39.9	12.6	10.0	39.3	26.4	94.6	87.8	0.15	0.13
Test	9.2	1.1	40.8	14.8	11.2	38.7	25.5	94.2	87.7	0.08	0.08
Validation	7.5	0.5	38.0	14.5	12.2	39.4	25.3	95.5	89.8	0.10	0.40

**Table 3 sensors-22-02758-t003:** Models’ parameters.

	Parameters	Layers
DeepLab	1,259,393	126
FC-DenseNet	685,032	130
DWT	38,147,984	46
Mask R-CNN	44,972,890	242
SA-UNet	2,914,910	26

**Table 4 sensors-22-02758-t004:** Training and inference times of the models.

	Training Time (Hours)	Inference Time (s)
DeepLab	3.8	0.06
FC-DenseNet	6.4	0.08
DWT	13	0.21
Mask R-CNN	8	0.23
SA-UNet	23	0.08

**Table 5 sensors-22-02758-t005:** Comparison of the mean IoU of each label achieved by each model.

	Eo	Ex	Es	Eso	Esx	Fao	Fax	Fbo	Fbx	Fco	Fcx
DeepLab	0	0	0.643	0	0	0.139	0	0.568	0.544	0	0
FC-DenseNet	0	0	0.701	0.340	0	0.519	0.332	0.630	0.534	0	0
DWT	0.157	0	0.670	0.278	0.289	0.481	0.323	0.725	0.693	0	0
Mask R-CNN	0.270	0.246	0.710	0.340	0.330	0.510	0.340	0.680	0.680	0	0
SA-UNet	0	0	0.722	0	0	0.489	0.080	0.745	0.710	0	0

**Table 6 sensors-22-02758-t006:** Comparison of model results.

	IoU of F2	Recall Rate	F-Score
SA-UNet	0.745	0.982	0.934
DNN [7]	0.680	-	-
DIAS-ResNet50-FPN [8]	-	0.905	0.931

## Data Availability

The data presented in this study are openly available on Kaggle.com. (accessed on 29 March 2022) The links are provided as follows. For the data: https://www.kaggle.com/changyuchi/ncu-ai-group-data-set-fcdensenet24 (Ionogram Data accessed on 5 August 2021).

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
