# Peer review of "State-of-the-Art Capability of Convolutional Neural Networks to Distinguish the Signal in the Ionosphere"

_sensors, 2022, doi:10.3390/s22072758_

Round 1
Reviewer 1 Report
Comments are in the attached file.

Author Response
Reply to Reviewer #1
We are grateful the Reviewer for very useful comments and suggestions. We addressed them in the revised manuscript.
- The specifications of used hardware and software are required to be mentioned in the results section.
The specifications are described in the beginning of Section 4. Results and Discussion:
“All of our models are written using Python 3.7.6 with NumPy 1.18.5, TensorFlow 2.3.0, and Keras 2.4.3., and the computations are conducted on Kaggle using two NVIDIA TESLA P100 GPUs and 16GB of RAM. The training and inference times of different models are listed in Table 4. It can be seen that the inference time is very short that makes it possible to apply the models for automatic processing of the ionograms in real time.”
- A table can be added in result section to compare the computational complexity (training, inference time, number of parameters) of all 5 CNN models, it can provide an insight for other researcher to select a specific model.
The training and inference times of different models are listed in a new Table 4.
- There is still no comparison with any other segmentation techniques. There are many publications in this area, so those benchmarks can be used to compare results with outcomes of other authors. Just comparison of CNN models is not enough for publication.
Yes, there are many various segmentation techniques as well as a lot of publications devoted to the image processing. Our paper is entitled “State of The Art Capability of Convolutional Neural Networks to Distinguish the Signal in The Ionosphere”. Hence, our work and results focus on CNN technique and its application for a particular task of ionogram recovery. In this study, we considered practically all previous results known in this particular area.
The application of another segmentation technique (with deeper learning) will be a subject of another ongoing paper for MDPI/Sensors.
- The source URLs should be moved to footnotes; they look strange in main text.
The footnotes are not allowed in MDPI papers.

Reviewer 2 Report
This paper classifies and compares five deep learning CNN models including SA-UNET , DWT , Mask RCNN , DeepLab and Fc-DenseNet24 based on the fact that ionospheric signals can be used to infer important physical properties of different layers of the ionosphere, and there are still some problems in the paper.
- the abstract section of the paper should state the final results, including the conclusions of the three factors of SNR, shape of signal and overlapping of signals.
- more references need to be cited in the paper, and the citations 16-25 are not standardized.
- As a research paper, this paper only discusses and compares some existing deep learning methods and experiments, and does not propose its own theory. The authors should improve at least one of the five types of deep learning methods and verify the effectiveness of the proposed methods.
Author Response
Reply to Reviewer #2
We thank the Reviewer for valuable comments and suggestions. The manuscript was revised accordingly.
- the abstract section of the paper should state the final results, including the conclusions of the three factors of SNR, shape of signal and overlapping of signals.
The abstract has been modified accordingly:
“Our results indicate that FC-DenseNet24, DWT, Mask R-CNN and SA-UNet are capable of identifying signals from very noisy ionograms (SNR< 1.4), overlapping signals can be well identified by DWT, Mask R-CNN and SA-UNet, and that more elongated signals are better identified by all models.”
- more references need to be cited in the paper, and the citations 16-25 are not standardized.
In this study, we considered previous results known in the quite new area of CNN technique application for image segmentation and, in particular, for a task of ionogram recovery. We will very appreciate the reviewer for any other relevant reference related to this topic.
- As a research paper, this paper only discusses and compares some existing deep learning methods and experiments, and does not propose its own theory. The authors should improve at least one of the five types of deep learning methods and verify the effectiveness of the proposed methods.
Our study is devoted to the experimental data treatment. The paper is entitled “State of The Art Capability of Convolutional Neural Networks to Distinguish the Signal in The Ionosphere”. Hence, it focuses on application and capability of CNN technique for a particular task of ionogram recovery. We do not aim to propose here any new theory/model. Nevertheless, we do have modified a few existing CNNs to improve their effectiveness of multi-label segmentation of ionograms. The detail is now described in Section 3.2 Models.
For example, FC-DenseNet was modified as the follows:
“The original version of FC-DenseNet, FC-DenseNet103 [10], consists of 670 layers and more than 19 million parameters. Given that our data set consists of more than 6000 images of 800x400 pixels, the computing power and memory space required to apply FC-DenseNet103 significantly exceed the limitation of our computing environment, Kaggle. Therefore, we constructed our lighter-weight version, FC-DenseNet24, by gradually reducing the number of layers and parameters until the model can run on Kaggle. The total number of layers and parameters are listed in Table 3, and the detail of the architecture is described in [1]. In both FC-DenseNet103 and FC-DenseNet24, the soft-max function is used as the activation function for the convolutional layer of output.”
The modification to Deep Waterched Transform, which was in original submitted version, is as follows:
“To allow multi-label assignment to a same pixel, we change the activation function of the output layer to sigmoid. Since the output probabilities are not normalized, they can be used in multi-label classification.”
The SA_Unet was modified in such a way:
“The original SA-UNet used 3x3 kernels in the convolutional blocks, and the DropRate was designed as ((1 - P) / (K2)) * (H / (H - K + 1)) * (W / (W - K + 1)), where K is the kernel size, H is the height of the image, W is the width of the image, and P is the keep probability. We changed the kernel size of the convolutional blocks to 7x7 to improve the overall performance, and modified the DropRate to be (1 - P) / (K2) to reduce the computational complexity.”

Reviewer 3 Report
Strong aspects:
The research compares the performance of several AI methods to extract useful information from ionograms.
Weak aspects:
The comparison with the results obtained in previous research is not clearly presented.
Comments to the authors:
- 'foF2' in line 59 is not explained.
- The abnormalities that should be detected by neural networks should be highlighted in Fig, 1 as in Fig. 3.
- In Table 2 the percentages are not clear. Typically the sum of the percentages for training, testing and validation should be 100%. I suggest presenting the total percentage for each characteristic (Eo, Ex, Es, Eso and so on) and independent percentage for train, test, and validation that with the sum 100 %.
Also, the authors should clarify why the sum on each row is more than 100 %.
- The results of the current research should be presented in a table with the results obtained in previous works. This will provide a clear view of the improvements brought by the current research.
- The equations (3) - (6) can be presented in the text because their complexity is low.
- The title of the last section 'Summary' should be replaced by conclusions.
Author Response
Reply to Reviewer #3
We appreciate the Reviewer’s Comments and Suggestions. They help us significantly to improve the quality of our paper.
- 'foF2' in line 59 is not explained.
Revised: “They reported an 85% success rate of determining foF2, which is the critical frequency (i.e., maximum frequency of the reflected signal) of the F2 layer in the ionosphere”
- The abnormalities that should be detected by neural networks should be highlighted in Fig, 1 as in Fig. 3.
Figure 1 has been revised accordingly.
- In Table 2 the percentages are not clear. Typically the sum of the percentages for training, testing and validation should be 100%. I suggest presenting the total percentage for each characteristic (Eo, Ex, Es, Eso and so on) and independent percentage for train, test, and validation that with the sum 100 %. Also, the authors should clarify why the sum on each row is more than 100 %.
This issue is clarified in the text of revised manuscript:
“Table 2 shows the percentage of sample numbers of each label in the training, validation and test sets. The percentage corresponds to the percentage of the ionograms that contain the indicated signal label. Since an ionogram can contain multiple labels and not all signal labels appear in all ionograms , the sum in each set is not 100%.”
- The results of the current research should be presented in a table with the results obtained in previous works. This will provide a clear view of the improvements brought by the current research.
For the comparison, we introduce a new Table 6:
“The best predictions of F2 from Mochalov & Mochalova (2019) [7] and Xiao et al. (2020) [8] are compared with our best prediction in Table 6.”
- The equations (3) - (6) can be presented in the text because their complexity is low.
The equations are moved in the text
- The title of the last section 'Summary' should be replaced by conclusions.
Replaced

Round 2
Reviewer 1 Report
Thanks, All my comments and suggestions have been addressed
Reviewer 2 Report
Thank the authors for their efforts. The authors have adequately addressed all my concerns in the review, and did a good job to revise and improve the paper. The paper now is suitable for publication in Sensors in its current form.
This manuscript is a resubmission of an earlier submission. The following is a list of the peer review reports and author responses from that submission.
Round 1
Reviewer 1 Report
This is a good contribution to the use of covolutional neural networks to interpreting and classifying ionosonde data. The paper is well written and deserves publication.
Author Response
Reply to Reviewer 1
We very appreciate the Reviewer’s time and efforts in evaluation of our work.
Comments and Suggestions for Authors
This is a good contribution to the use of covolutional neural networks to interpreting and classifying ionosonde data. The paper is well written and deserves publication.

Reviewer 2 Report
In the paper 5 existing architectures of convolutional neural networks were used for segmentation of ionograms. There are several problems, however, that should be addressed:
- Authors mention about alternative automatic segmentation techniques (e.g. fuzzy segmentation) but they do not compare their results with them. Without that It is hard to objectively assess the quality of the presented results.
- If there are other publications in this area then perhaps there are some other benchmarks which would allow to compare results with outcomes of other authors. It was not discussed in the paper.
- As I understand the considered problem is a multilabel segmentation problem (overlapping of signals). Typically the used CNN architectures assign one label to a single pixel. It was not discussed how this problem was solved at all.
- Authors wrote about validation set which was used for supervising and calibrating. Typically validation set is used for model and hyperparameter selection to avoid underfitting and overfitting problems. Those problems were not discussed (the usage of validation set is not in the paper).
There are also several problems with presentation:
- In the introduction there is only a short mention about alternative automatic segmentation techniques. I would expect the whole related works section here.
- In my opinion authors should not write about convolutional models (section 3.2), in particular taking figures from other papers. Those models are well known and it is enough to cite them only. It would be beneficial if details of their application (modifications if there are any) for the specific was presented.
- In the whole paper there is no sample result of manual (ground truth) and automatic (network outputs) segmentation. It would allow reader better understand the considered problem.
- There are 11 classes but in results only 3 of them are discussed.
Other comments:
- File name from Figure 1 should be removed and there is no unit description on horizontal axis.
- URLs look strange in the text, I would consider moving them to footnotes.
- Equations are not numbered.
- In line 166 there is k symbol which was not explained (it is used as an index and as an upper limit of sum operator).
- All the equations should sections 3.3.1-3.3.3 should be described with more details.
Author Response
Reply to Reviewer 2
We are very grateful to the Reviewer for the useful comments and suggestions. We have revised the manuscript accordingly.
Comments and Suggestions for Authors
In the paper existing architectures of convolutional neural networks were used for segmentation of ionograms. There are several problems, however, that should be addressed:
Authors mention about alternative automatic segmentation techniques (e.g. fuzzy segmentation) but they do not compare their results with them. Without that It is hard to objectively assess the quality of the presented results.
It is hard to make numerical estimation of the fuzzy technique accuracy because it is unsupervised. Hence the formalism of IoU (as well as others) cannot be used for comparison of the fuzzy segmentation with the CNN results.
If there are other publications in this area then perhaps there are some other benchmarks which would allow to compare results with outcomes of other authors. It was not discussed in the paper.
We add several references and revised Introduction accordingly:
“At present, Convolutional Neural Networks (CNNs) have been applied in automatic ionogram signal recovery [15]. In their work, they have successfully recovered the ionospheric signal from the F-layer from experimental ionogram data in Peru. Their results show that their model can recover F-layer signals with an average Intersection over Union (IoU) of 0.569. Another work also applied CNN for ionogram recognition using ionosonde data from “Parus-A” which was operated by the Institute of Cosmophysical Research and Radio Wave Propagation [16]. In their work, they used a deep learning model known as U-Net. However, they do not provide numerical estimation of the model accuracy.”
As I understand the considered problem is a multilabel segmentation problem (overlapping of signals). Typically the used CNN architectures assign one label to a single pixel. It was not discussed how this problem was solved at all.
Actually some advanced CNN models enable multi-label segmentation with overlapping. In order to demonstrate it, we have modified Figure 5 in the revised manuscript.
“Figure 5 shows an example of the effect of overlapping. Panel (a) is the original ionogram which contains the signals Fbo and Fbx, and panel (b) is the ground truth, i.e., the labeled signals. The overlapping region is marked in red. Fbo and Fbx identified by the model DeepLab are plotted in panels (c) and (d), and those identified by the model FC-DenseNet24 in panels (e) and (f). Panels (c) and (d) show that identified Fbo and Fbx are both continuous with almost no gaps, which means that most of the pixels in the overlapping region are correctly identified as both Fbo and Fbx by DeepLab. In contrast, a wide gap can be seen in panel (f). This indicates that the model FC-DenseNet24 is less capable of identifying the two signals in their overlapping region. Further, we will examine whether and by how much OR would affect the prediction accuracy of different models.”
Authors wrote about validation set which was used for supervising and calibrating. Typically validation set is used for model and hyperparameter selection to avoid underfitting and overfitting problems. Those problems were not discussed (the usage of validation set is not in the paper).
Results for the validation set are presented in Table 2.
We also improved the description of the meaning of validation set in according to the Reviewer’s suggestion (see lines 112-114):
“The validation set is used for supervising and calibrating the model during the training, and for the selection of hyperparameters to avoid underfitting and overfitting problems, and the test set is used to test the performance of the model after the whole training.”
There are also several problems with presentation:
In the introduction there is only a short mention about alternative automatic segmentation techniques. I would expect the whole related works section here.
This subject is quiet new. Hence, there are not so many publications devoted to segmentation of ionograms. We will very appreciate the Reviewer for recommendation of any relevant paper. In Introduction of the revised manuscript, we have added several references related to the applications of CNN networks for the image segmentation:
- De la Jara Sanchez, C. Ionospheric Echoes Detection in Digital Ionograms Using Convolutional Neural Networks. Ph.D. Thesis,Pontificia Universidad Catolica del Peru-CENTRUM Catolica, Surco, Peru, 2019, doi: 10.1029/2020RS007258.
- Mochalov, Vladimir, and Anastasia Mochalova. Application of deep learning to recognize ionograms. In 2019 Russian Open Conference on Radio Wave Propagation (RWP), Tatarstan, Russia, 1 – 6 Jul. 2019, doi: 10.1109/RWP.2019.8810326.
- Xiao, Z., Wang, J., Li, J., Zhao, B., Hu, L., & Liu, L. Deep-learning for ionogram automatic scaling. Advances in Space Research 2020, 66.4: 942-950, doi: 10.1016/j.asr.2020.05.009.
- Furukawa, Ryouichi, and Kazuhiro Hotta. R. Localized Feature Aggregation Module for Semantic Segmentation. In 2021 IEEE International Conference on Systems, Man, and Cybernetics (SMC), Melbourne, Australia, 17 – 21 Oct. 2021, doi: 10.1007/s10489-021-02603-z.
- Quan, B., Liu, B., Fu, D., Chen, H., & Liu, X. Improved Deeplabv3 For Better Road Segmentation In Remote Sensing Images. In 2021 International Conference on Computer Engineering and Artificial Intelligence (ICCEAI), Hebei, China, 22 – 24 Jul. 2022, doi: 10.1109/ICCEAI52939.2021.00066.
- Niu, Z., Liu, W., Zhao, J., & Jiang, G. DeepLab-based spatial feature extraction for hyperspectral image classification. IEEE Geoscience and Remote Sensing Letters 2018, 16.2: 251-255, doi: 10.3390/rs12091395.
- Hai, J., Qiao, K., Chen, J., Tan, H., Xu, J., Zeng, L., ... & Yan, B. Fully convolutional densenet with multiscale context for automated breast tumor segmentation. Journal of healthcare engineering 2019, doi: 10.1155/2019/8415485.
- Guo, Xuejun, Zehua Chen, and Chengyi Wang. Fully convolutional DenseNet with adversarial training for semantic segmentation of high-resolution remote sensing images. Journal of Applied Remote Sensing 2021, doi: 10.1117/1.JRS.15.016520.
- Li, Y. H., Putri, W. R., Aslam, M. S., & Chang, C. C. Robust Iris Segmentation Algorithm in Non-Cooperative Environments Using Interleaved Residual U-Net. Sensors 2021, 21.4: 1434, doi: 10.3390/s21041434.
In my opinion authors should not write about convolutional models (section 3.2), in particular taking figures from other papers. Those models are well known and it is enough to cite them only. It would be beneficial if details of their application (modifications if there are any) for the specific was presented.
We have removed Figure 5 representing DeepLab. Other Figures are original and they demonstrate the key modifications of the initial networks for the particular task of ionogram image segmentation. The modifications are described in the text.
In the whole paper there is no sample result of manual (ground truth) and automatic (network outputs) segmentation. It would allow reader better understand the considered problem.
In the revised manuscript, we have modified Figure 5, which now shows “An example of the overlapping ordinary and extraordinary signals from the F2 layer: (a) the original ionogram; (b) the ground truth labeled in the original ionogram (black pixels are the non-overlapping part and red pixels the overlapping part of the signals); (c) and (d) the output of DeepLab for Fbo and Fbx, respectively; (e) and (f) the output of FC-DenseNet for Fbo and Fbx, respectively.”
There are 11 classes but in results only 3 of them are discussed.
We add the following clarification at lines 255-258:
“As one can see in Table 4, only three classes are predicted by all the models with high accuracy of IoU>0.6: Es, Fbo, and Fbx. In the next subsections, we will use these three classes to analyze how various effects influence the accuracy of the five models considered.”
Other comments:
File name from Figure 1 should be removed and there is no unit description on horizontal axis.
Figure 1 have been revised accordingly.
URLs look strange in the text, I would consider moving them to footnotes.
The footnotes are not allowed in the MDPI papers.
Equations are not numbered.
Corrected
In line 166 there is k symbol which was not explained (it is used as an index and as an upper limit of sum operator).
Corrected:
“where k is the number of ionograms analysed. It should be noted that the value defined by equation (1) is calculated for the whole ensemble of ionograms for a given set (training, validations and testing).”
All the equations should sections 3.3.1-3.3.3 should be described with more details.
Subsection 3.3.1: (SNR) has been enhanced substantially in order to describe the definition of SNR with more details.
Subsection 3.3.2: There are no equation. The definition of C/A is very complicated. This characteristic is introduced in [1].
Subsection 3.3.3: We clarify the Equation (6) as the following:
“where signal1 is the area of the signal of interest and signal2 is the area of another signal, which intersects the signal of interest. In this work, we focus on the overlapping situation of the F2 layer because Fbo and Fbx usually have larger overlapping areas than other layers.”

Reviewer 3 Report
State of The Art Capability of Convolutional Neural Networks to Distinguish the Signal in The Ionosphere
In this paper, the authors proposed the Convolutional Neural Networks based models for identification of ionospheric signals. A comparative analysis of different deep learning models is performed by the authors and SA-UNet is claimed to be the best among all to identify these signals. This is an interesting research topic and accurate identifications of ionospheric signals can be useful for communication, navigation, climate studies and positioning applications.
The paper under discussion is well-written and easy to understand. Language of paper is technical and need no further changes. Introductory part is well defined. However, following comments need to be addressed:
- The novelty of the proposed scheme is really limited. A lot of other researchers have done work on same approach of using CNN models for this specific task, so, nothing new contributes to the state of the art.
- For example, Mendoza, M.M.; Chang, Y.-C.; Dmitriev, A.V.; Lin, C.-H.; Tsai, L.-C.; Li, Y.-H.; Hsieh, M.-C.; Hsu, H.-W.; Huang, G.-H.; Lin, Y.-C.; et al, “Recovery of Ionospheric Signals using Fully Convolutional DenseNet and Its Challenges,” Sensors 2021, 21, 6482.
- The models proposed in this paper needs to be compared with other state of the art networks proposed by others.
- As a well-researched area, the number of latest reference in the manuscript seem to be weak. Some other good and recent references need to be added.
- The conclusion section is fine. Please add the future direction part. etc.
Author Response
Reply to Reviewer 3
We thank the Reviewer for valuable comments and suggestions. We have revised the manuscript in according to the suggestions.
Comments and Suggestions for Authors
State of The Art Capability of Convolutional Neural Networks to Distinguish the Signal in The Ionosphere
In this paper, the authors proposed the Convolutional Neural Networks based models for identification of ionospheric signals. A comparative analysis of different deep learning models is performed by the authors and SA-UNet is claimed to be the best among all to identify these signals. This is an interesting research topic and accurate identifications of ionospheric signals can be useful for communication, navigation, climate studies and positioning applications.
The paper under discussion is well-written and easy to understand. Language of paper is technical and need no further changes. Introductory part is well defined. However, following comments need to be addressed:
The novelty of the proposed scheme is really limited. A lot of other researchers have done work on same approach of using CNN models for this specific task, so, nothing new contributes to the state of the art.
For example, Mendoza, M.M.; Chang, Y.-C.; Dmitriev, A.V.; Lin, C.-H.; Tsai, L.-C.; Li, Y.-H.; Hsieh, M.-C.; Hsu, H.-W.; Huang, G.-H.; Lin, Y.-C.; et al, "Recovery of Ionospheric Signals using Fully Convolutional DenseNet and Its Challenges," Sensors 2021, 21, 6482.
The present paper continues the paper by Mendoza et al., 2021, where we presented the data set of ionograms, described it main properties, and made a first attempt to model it with using a CNN. A reviewer of Mendoza et al., 2021 asked us to demonstrate more models for ionogram segmentation. The purpose of this paper is determined in the Introduction:
“In this work, the aim is employing advanced algorithms of deep learning to automatically classify each layer. Specifically, we train and apply five artificial neural network models: DeepLab [5], Fully Convolutional DenseNet (FC-DenseNet) [1, 6], Deep Water-shed Transform (DWT) [7, 8], Mask R-CNN [9], and Spatial Attention U-Net (SA-UNet) [10, 11], all of which have been successfully applied to image segmentation. It is important to note that all these models are available for everybody at https://www.kaggle.com/ and, thus, can be easily applied and verified.”
The models proposed in this paper needs to be compared with other state of the art networks proposed by others.
We would note that all the models used in the paper are available for everybody at https://www.kaggle.com/ and, thus, can be easily verified. The access to more advanced artificial neural networks (ANNs) is difficult. Moreover, those ANNs require more advanced computing facilities. This task is beyond of the subject of current paper. We are preparing a separate paper devoted to really advanced ANN. We will also appreciate the Reviewer for access (if possible) of such kind of advanced ANN and computing facilities.
As a well-researched area, the number of latest reference in the manuscript seem to be weak. Some other good and recent references need to be added.
We have added more references related to CNN model technique and its application in Introduction (see above), and in Summary (see our reply to the next comment)
The conclusion section is fine. Please add the future direction part. etc.
We have added one more paragraph in the end of Summary:
“We have shown that the CNN models presented provide high accuracy only for three classes: Es, Fbo, and Fbx (see Table 4). The other classes are identified by the models with very poor accuracy or even cannot be predicted at all. We believe that a more advanced network with deeper learning [24-26] will give better accuracy in segmentation of the ionograms. The development of advanced models will be a subject of our future work.”

Round 2
Reviewer 2 Report
The paper has been improved, however, several main issues are still there:
- There is still no comparison with any other segmentation technique. Authors wrote in their reply that: "It is hard to make numerical estimation of the fuzzy technique accuracy because it is unsupervised". Unsupervised character of method has no influence on results as segmentation is segmentation and only outputs must be compared with ground truth. As I understand the goal of the work is to show that CNNs are good tool for such segmentation. But it makes no sense to use them if the other techniques can do that better. Without comparison the hypothesis cannot be proved.
- Authors wrote: "Other Figures are original and they demonstrate the key modifications of the initial networks for the particular task of ionogram image segmentation. The modifications are described in the text. ". In my opinion that is not true. In CNN architectures descriptions there is no information about their adaptation to ionogram segmentation. That is why section 3.2 describing the existing works is not a good idea. Below I describe how it can be replaced.
- Authors claim that the used CNN architectures can be used in a natural way for multi-label segmentation. In my opinion it is not that obvious and should be clarified. Typical CNNs used for segmentation has a pixel classifier at the end usually with softmax layer. This layer gives us the probabilities of all labels for every pixel (the label with highest probability is assigned to that pixel). It is not clear how authors achieve the effect show in Figure 5 where some pixels have two labels. Do they select two highest probabilities (if so why only two and was the threshold)? Or maybe they train several binary classifiers for all the labels? It must be explained.
- My comment about validation set should not lead to Table 2 extension. Authors wrote: "The validation set is used for supervising and calibrating the model during the training, and for the selection of hyperparameters to avoid underfitting and overfitting problems". What kind of hyperparameters were cllibrated and consequently how the uderfitting and overfitting problems were overcome?
Other comments:
- Mathematical notation still is not precise. For example, symbol k was explained in equation (1) was explained but they use it both as an index and as an upper bound of summation. Please check it carefully once again.
Reviewer 3 Report
I still think that the paper needs some major modifications to be considered for publication. Paper should be more structured and should show the novelty of your work in a clear way. A comparison of performance with other works would be good to prove the superiority of proposed algorithm.